# Islam and the Politics of Secularism in Pakistan

Zahid Shahab Ahmed 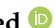

Alfred Deakin Institute for Citizenship and Globalisation, Deakin University, Burwood, VIC 3125, Australia; zahid.ahmed@deakin.edu.au

**Abstract:** In terms of their political and ideological success, Pakistani Islamists have had several ups and downs since Pakistan became the Islamic Republic in 1956. Islamists strive to safeguard the Islamic state's status quo while simultaneously expanding the reach of Sharia. Despite insignificant electoral victories, Islamists have largely been able to dictate national identity policies to civilian and military governments. A major hurdle to the promotion of pluralism in Pakistan is noticeable through persistent opposition to secularism by major political actors. Despite different political ideologies, major political parties refrain from promoting secularism in Pakistan; however, such views are more rigid in the case of Islamists. The purpose of this article, therefore, is to examine and compare the discourses of Islamists and other political parties in relation to Pakistan's identity, reforms and anti-Westernism, religious minorities, and secularism. Based on the analysis, this paper argues that the views of Islamists and non-religious political parties are very similar regarding Islam and Pakistan's identity, secularism, and minority rights in Pakistan.

**Keywords:** Islam; domestic politics; Islamists; Pakistan; secularism

## 1. Introduction

Since becoming the Islamic Republic of Pakistan in 1956, Islam has been revered as the supreme authority in Pakistan. The country, however, has struggled to form a singular identity based on the dominant religion (Islam) because of ethnic divisions. Religious identity also raises questions regarding the inclusion of religious minorities in the country. Despite having a 96.47 percent Muslim population, Pakistan contains approximately four percent of religious minorities, such as Christians, Hindus, Ahmadiyyas, Parsis, Buddhists, Sikhs, and others (PBS 2017). The 2017 census is controversial for a couple of reasons. First, the data on religious minorities were shared nearly two years after the census was released (Tunio 2019), and second, the Christian community has criticized the census for under-reporting their numbers in Pakistan (N. Hoodbhoy 2021). In terms of the percentage of the total, the share of religious minorities might appear small, but that is not the case if we look in numbers as there are approximately 7 million non-Muslims in Pakistan's 207 million population (PBS 2017). These demographics have changed in relation to the proportion of religious minorities in Pakistan since 1947. At the time of the creation of Pakistan, there were approximately 23 percent non-Muslims in Pakistan, including the majority in the East Pakistan (Ispahani 2013). The demographics called for pluralism, which was exemplified by the founding father Muhammad Ali Jinnah, for example, by the appointment of Chaudhry Zafarullah Khan (Ahmadiyya) as foreign minister and Jogendra Nath Mandal (Hindu) as law minister in his first cabinet (P. Hoodbhoy 2016, p. 36). Another example of pluralism in the vision of Pakistan's founders is an amendment to the country's flag, in which a white strip was added to symbolize the nation's religious minorities (Burki 1986, p. 43). It should have been clear what Jinnah's plan for Pakistan was after the creation of such a diverse government and the flag, but that was not the case because Jinnah died in 1948, taking his hard-won vision with him. Soon after his death, the debates surrounding Pakistan's Islamic identity dominated, and the country was declared an Islamic Republic. In this backdrop, this paper aims to answer the following questions: How has the secularism

debate in Pakistan evolved since 1947? How have various actors perceived the ideas of an Islamic State or a Muslim state? How do various political actors understand secularism and promote or oppose it?

Thinking of secularism, one would tend to view many Western countries as more secular compared to non-Western states because Western states are known for protecting and promoting freedom of religion and belief (Possamai and Possamai-Inesedy 2021). In its true sense, secularism is about separating religion from the state (Ahmar 2012; Iqtidar and Gilmartin 2011), but does that really happen, even in the case of renowned secular states? As argued by Asad (2003), there is a pattern demonstrating the state's involvement in religious affairs by managing religious practices and thought. It has, however, not been easy in many countries. Sometimes the secular dilemma leads to the politicization of religion, as was the case in Lebanon (Andersen 2022). Then, not all secular states are alike, for instance, in terms of separating religious institutions and the state, as Nweke (2015, p. 85) argues, "although more feasible within secular states, religious freedom and human rights are possibilities relative to existential circumstances of states, irrespective of their secular or non secular leanings". As secularism has progressed, albeit differently, across the Muslim world, Muslims scholars have focused on the compatibility between Islam and secularism. There is, however, an overwhelming majority that has viewed secularism as a social evil. Ashimi (2022, p. 55) claims, "Indeed, with the mindset of separating religion from worldly life, secularism gives a negative impact on the morality of humans and turns them to the state of ignorance". Similarly, AbdelSalam (2004, p. 216) argues, "Contrary to the claim of many Western writers, secularism cannot be considered a prerequisite for successful operation of democracy in Muslim societies, nor can it be considered a requirement of modernity. False modernity manifests itself in the deconstruction of Islamic society and then rebuilding it on a new non-Islamic basis". It is not surprising to see many Muslim writers and scholars rejecting secularism as theologically Islam rejects secularism or the separation of state and religion (Jackson 2017).

Each non-Western state has a different journey, and such is the case in Pakistan, which was formed for the Muslims of the Indian Sub-continent in 1947. Hence, Islam has been holding a central stage in Pakistan, but there have always been divisions between agents of Islamization and secularism. In 1956, Pakistan was declared an Islamic Republic, and its constitution said that it adhered "to a diluted version of pluralism" (P. Hoodbhoy 2016, p. 36). During Liaqat Ali Khan's tenure as Pakistan's first prime minister, the "Objective Resolution" was adopted, and that paved the way for the creation of the Islamic Republic of Pakistan. Under the military rule of Field Marshal Ayub Khan, Pakistan underwent various reforms as Khan wanted Pakistan to be a modern Islamic state but a secular one, as was also reflected in the 1963 constitution of Pakistan (Qasmi 2019; Chacko 2016). He embarked on a nation-building project by re-writing history and claimed secularism (Qasmi 2019), but the reality was different. Khan gave in to the demands of the mullahs, and the Central Institute of Islamic Research was established in 1959 (Chacko 2016). Although the Objective Resolution hinted at the government's apparent affiliation with Islam, Sharia was not mentioned in the document (Zaidi 2003, p. 102), which happened later on while Pakistan experienced Islamization by Zulfiqar Ali Bhutto in the 1970s. The 1973 constitution made Pakistan a theoretical state in every possible manner as Islam became the state religion and only a Muslim could be the head of state (Hamdani 2022). Article 2 of the constitution of Pakistan declared Islam as the state religion: "Islam shall be the State religion of Pakistan" (GOP 2012, p. 3). Even after the approval of the constitution, religious groups—led by the Jamaat-e-Islami—continued their pressure on the government to declare Ahmadiyya non-Muslims. Bhutto was against the idea of discussing the Ahmadiyya issue in the parliament and "maintained that declaring the Ahmadiyya a minority and pushing them out from state and government institutions would be detrimental to the economy of the country" (N. F. Paracha 2013).

Islamization resulted in the adoption of strict Islamic laws during the Zia-ul-Haq regime in the 1980s. A study argues that Zia-ul-Haq used Islam to support his regime's

"survival strategies of legitimacy, repression, and social control" (Sheikh and Ahmed 2020, p. 333). The legal reforms included the Hudood Ordinance and the murky Blasphemy Law. The media also had a period of intense state supervision, during which time television drama was to suffer a significant setback with the return of Islamic fundamentalism, and the state-run television channel (Pakistan Television or PTV) was utilized to disseminate both political and religious messages (Burki 1986, p. 33). Additionally, the Ministry of Information ordered PTV's female newsreaders to appear on television without makeup and with a *dupatta* (scarf) covering their heads. In defense of Islamization, Zia-ul-Haq said, "Pakistan is like Israel, which is an ideological state. Take out the Judaism from Israel and it will fall like a house of cards. Take Islam out of Pakistan and make it a secular state; it would collapse" (Burki 1986, p. 78). While he was motivated by Saudi Arabia's Islamic principles, structural changes were taking place in the nation because of Saudi Arabia's strategic alliance with the US in relation to the Afghan–Soviet War (1979–1988). Then, for their political purposes, Jamaat-e-Islami (JI) and Jamiat Ulema-e-Islam (JUI) were at the forefront of jihad and Islamization both domestically and overseas (Afghanistan) (Z. S. Ahmed 2012).

Regarding the erstwhile scholarship on Islam and politics in Pakistan, there is no shortage in the literature in terms of how various political actors have used Islam in domestic politics and how that has led to a gradual decline in secularism in the country. In 2011, a Special Issue of *Modern Asian Studies* was dedicated to secularism in Pakistan in which the editors argued, "The case of Pakistan is an excellent for approaching secularism in this way, for it provides a clear perspective on how the analysis of secularism cannot be limited by any simple formula for separation of religious from the state" (Iqtidar and Gilmartin 2011, p. 494). I. Ahmed (2004) presented a comprehensive analysis of historical accounts of secularism and debates surrounding Islam and secularism in Pakistan. Jaffrelot (2012) examined Pakistan's trajectory in terms of Islamization and how the country moved away from the secular vision of Muhammad Ali Jinnah. Similarly, Ahmar (2012) focused on Islamization in Pakistan and the two different camps for and against secularism. There is, however, no previous study that has examined and compared the relevant discourses of major religious and non-religious political parties in the country. This study aims to bridge that gap through a critical discourse analysis of how Islamists and non-religious political parties have viewed the role of Islam in Pakistan.

There has been no prior study that has investigated how major political parties, including both those that claim to be secular and Islamists, perceive the concept of secularism and what their visions/manifestos are in relation to secularism, pluralism, and the status of religious minorities in Pakistan. This paper aims to bridge that gap by investigating and comparing the positions (discourse) of the so-called seculars, i.e., Pakistan Muslim League-Nawaz (PML-N), Pakistan Peoples Party (PPP), and Pakistan Tehrik-e-Insaf (PTI), and Islamists including JI and the Jamiat Ulema-e-Islam Fazl (JUI-F). Here, the author has categorized JI and JUI-F as Islamists because both parties support Islamic fundamentalism and the greater influence of Islamic law in the politics and society of Pakistan. Critical discourse analysis (CDA) is used as an analytical tool to examine the relevant discourse and policies promoted by the selected political parties. CDA was previously used to examine the discursive construction of national identities by states (Alameda Hernández 2008; Blommaert and Bulcaen 2000), but it cannot be disconnected from the context as "the genre of language is associated with a particular social activity" (Fairclough 1993, p. 100). For the analysis, the data were collected from primary and secondary sources, such as the websites of political parties and their manifestos, and newspaper reports, covering a period of 20 years between 2002 and 2022. This period is quite relevant because the 2002 general elections were organized under the military rule of General Pervez Musharraf, in which Islamists through the Muttahida Majlis-e-Amal (MMA) achieved significant success by forming a government in Khyber Pakhtunkhwa (KP) during 2002–2007. The PPP and its allies formed a government following the 2008 elections, the PML-N after the 2013 elections, and the PTI because of the 2018 elections. As most of the parties are old, the analysis uses

historical references while talking about the ideas of party leaders such as Maulana Sayyid Abul A'la Maududi of the JI. The analysis is divided into two sections separately examining the discourse of Islamists and non-religious parties on the following themes: (1) Pakistan's identity; (2) reforms and anti-Westernism; (3) religious minorities; and (4) secularism.

## 2. Secularism in Pakistan: Islamists versus Seculars

In this section, a comparative analysis is carried out on the positions of Islamists and seculars on issues relevant to secularism in Pakistan.

### 2.1. Pakistan's Identity

Concerning Pakistan's status as an Islamic state, there has always been ambiguity. Concerns have been raised about the place of secular ideals in a society where religion predominates. Since Pakistan's formation, there have been differences concerning the national identity of Pakistan and how it was perceived by the country's founders. In fact, there are opposing viewpoints regarding Muhammad Ali Jinnah's vision for the nation; for instance, whether he would have preferred an Islamic Republic or a secular state. This dispute occasionally flares up and was very active during the years this study was conducted.

Islamists continue to oppose secularism based on their own vague understanding of it. Islamists connect secularism as incompatible with the fundamental Islamic teachings. They are unwilling to make any concessions to allow for the promotion of plurality in the country since their viewpoint is so rigid. Later in the paper, numerous examples are provided of how and when Islamists have used the label of secularism to target non-religious and dominant political parties. Here, however, it is important to mention that they are not the only ones confused about secularism. As Ahmar (2012, p. 217) argues, "secularism is perceived as the most misinterpreted term in Pakistan". Across the board, Islamists in general, and particularly, the leadership of JI and JUI-F, have been claiming that Pakistan was created to be an Islamic state in which the Muslims of the Indian sub-continent could live under Sharia. To justify their stances, they provide references to the blueprint of Pakistan or the Two-Nation Theory. For example, see what JUI-F's Amjad Khan once said regarding the identity of Pakistan:

> Pakistan was created for Muslims and Islam, but over the past 70 years it has been led astray by Western conspiracies and secular forces . . . The MMA's reunion, which we have been trying to achieve for some time, is to establish true Islamic values and create the Pakistan that was envisions by the founding fathers as per the Two-Nation Theory. (K. K. Shahid 2018)

To justify their positions regarding Pakistan being the Islamic Republic, Islamists continue to support the Two-Nation Theory. Leaders of JI and JUI-F often refer to Allama Muhammad Iqbal—credited for developing the Two-Nation Theory—in their speeches. This is despite the fact that the founders of both parties were against the creation of a separate country for the Muslims of the Indian sub-continent (Jaffrelot 2012). As Tohid (2003) argues, "While the religious parties were opposed to the creation of Pakistan, they have cashed in on the two-nation theory, arguing that the country was formed on the basis of the religious divide". JI leaders have been trying to promote similarities in the visions of Maududi and Iqbal in terms of the revival of Muslims. While addressing a conference on "Iqbal and Maududi's Pakistan" in 2019, JI chief Sirajul Haq said, "Dr Iqbal had a vision of an independent state for the Muslims of the Subcontinent and Maulana Maududi had worked hard in teaching them the lessons of Holy Quran and Sunnah" (The News 2019). Haq earlier had tweeted on this issue in 2017 on the Independence Day of Pakistan (14 August): "Pakistan was created on the basis of Two nation theory. We should work together to implement Islam teachings in true letter and spirit" (S. u. Haq 2017). In contrast, JUI-F leaders have spoken little about the Two-Nation Theory as their narrative has mainly focused on targeting non-religious parties and the West. For instance, the chief of JUI-F, Maulana Fazl-ur-Rahman, said in December 2022 that "international anti-Islam

forces are hatching conspiracies to politically and economically destabilize Pakistan . . . the agents of those forces were making efforts to weaken . . . the Constitution" (S. Shahid 2022). For Islamists, the Two-Nation Theory provides a justification for the Islamic Republic of Pakistan or an Islamic state.

The views of non-religious parties are not different from religious parties when it comes to the supremacy of Islam in Pakistan or the very fact that the country was created for the Muslims of the Indian sub-continent. From Bhutto's Islamic socialism to Imran Khan's vision of a model Islamic state (Riasat-e-Medina), there is enough to suggest that non-religious parties are also in favor of a theocratic state. Like the leaders of religious parties, leaders of most non-religious parties also refer to the Two-Nation theory and quote Muhammad Ali Jinnah to present their interpretations of the state identity, which are quite identical. There was, however, an occasion when Nawaz Sharif of PML-N came under attack following his speech to an audience comprising Sikh guests from India. Addressing the guests, he said, "many attributes of culture, including places of origin, food and dress, and a common Provider (Rab)" between Islam and Sikhism (Rehman 2011). While he made no mention of the Two-Nation Theory, religious groups such as the JI viewed his remarks as contradictory to the vision of Pakistan or the Two-Nation theory (Rehman 2011). During the PTI government (2018–2021), several party members made references to the Two-Nation Theory to criticize the Indian government's targeting of Muslims. In addition to Imran Khan, Dr Firdous Ashiq Awan said, in her capacity as the Special Assistant to Prime Minister on Information and Broadcasting, that the "Two-Nation Theory of Quaid-i-Azam Muhammad Ali Jinnah had won in India and how Indians were remembering the heroes of the theory" (Wakeel 2019). The PPP is different in this regard because its leaders carefully associate Islamic identity with Pakistan. This could be because of their minority vote bank and because of the secular orientation of its key leaders; for example, Bilawal Bhutto, tweeted in 2020:

> Quaid-e-Azam Muhammad Ali Jinnah August 11th 1947 promise: "you are free to go to your temples, you are free to go to your mosques or to any other place of worship in this State of Pakistan. You may belong to any religion . . . has nothing to do with the business of the State. (Zardari 2020)

There is a long history of parties moving from left to right in terms of their ideologies, and this could be because they find it hard to question the very foundation of the state, i.e., religion based on the Two-Nation Theory. Let us take the example of PPP's Zulfiqar Ali Bhutto and his ideals of Islamic socialism. In addition to aiming to eradicate feudalism and unregulated capitalism, Bhutto's Islamic socialism "denounced the conservative religious parties and the clergy of being representatives of monopolist capitalist, feudal lords, military dictators, 'the imperialist forces of capitalism,' and being agents of backwardness and social and spiritual stagnation" (N. N. Paracha 2013). The JI was able to persuade hundreds of Islamic scholars (*ulema*) to label the PPP as "atheistic" and "anti-Islam" which eventually led to a split/conflict between the PPP and religious parties (N. N. Paracha 2013).

In terms of the relationship between the state and religion, PTI moved to the next level as its chief, Imran Khan, has been calling for creating a "Riasat-e-Medina" in Pakistan. According to Surbuland (2022, p. 5)., "this is a powerful piece of rhetoric which claims to emulate the social-political values of the Prophet Muhammad in the present day"—exactly what the JI's Maududi preached Here again, a gap in rhetoric and reality is noticeable as Khan has repeatedly praised the Western political institutions and good governance in China that has led to a large-scale poverty alleviation (Surbuland 2022; Standish and Khattak 2021). Khan's ideals are not very different from Bhutto's Islamic socialism. Khan benefited by emphasizing the idea and rhetoric of "Islamic socialism" to establish legitimacy in the eyes of Pakistan's Muslim people. The poor and working class of Karachi were the most ardent new supporters, as evidenced by a journalist's report from October 2018 (Judah 2018). Khan also used his emphasis on the idea of "Islamic socialism" to disparage the practices carried out by earlier governments. Because the underprivileged in Pakistan have

had "no safety other than their own families or tribes," according to Khan, the country has never genuinely been an Islamic state (Campbell 2018).

*2.2. Reforms and Anti-Westernism*

A prominent feature of the selected Islamists is noticeable through their outright opposition to certain reforms concerning security and women's rights and how they have justified their positions by linking their rhetoric with secularism. In 2014, JI's Liaqat Baloch called Pakistan's security reforms regarding counterterrorism "a conspiracy to turn Pakistan into a secular country" (Dawn 2014). Baloch mentioned madrassah reforms in Pakistan that were mainly introduced following the start of the "War on Terror", and he used such examples to target the PML-N government of that time. He said, "the secular lobby in the country was out to shake the foundations of the mosques and the madrassahs to fulfill a western agenda" (Dawn 2014). In March 2016, JUI's chief Maulana Fazl-ur-Rahman considered the Punjab Government's Protection of Women Against Violence Bill against Sharia and declared that he would oppose it (Ali 2016). The religious parties unanimously opposed the women's rights bill by deeming it a conspiracy "to make Pakistan a secular country" (Ali 2016). This kind of anti-secular discourse is frequently employed by religious parties as a political weapon against governments.

The targeting of other self-proclaimed secular or non-religious parties is a key feature of Pakistani Islamists. This anti-secular discourse is frequently employed by religious parties as a political weapon against governments. For instance, Sirajul Haq, the leader of the JI, criticized the PML-N administration by claiming that "the Prime Minister's slogan of a liberal and secular Pakistan and PPP leader Bilawal Bhutto's move for a political alliance against the religious parties was an indication that whereas the Qibla of the rulers was the US and Washington" (S. Haq 2016a). Even in the Panama Leaks corruption scandal, Sirajul Haq offered a religious viewpoint: "only liberal and secular people were named by the Panama Leaks for tax evasion and there was not a single religious scholar or [madrassa] head facing corruption charges" (S. Haq 2016b). Earlier during the Musharraf era, JI was openly against various reforms that Islamists in general and the JI in particular had viewed as secular.

The opposition of Pakistani Islamists to secularism is strongly linked to their historic anti-Westernism. In addition to often delivering anti-Western speeches, Islamists have also been engaged in rallies against Western countries. Following the publication of blasphemous cartoons by Charlie Hebdo in 2015, hundreds of protestors from the JI and JUI-F protested across Pakistan. In response, the Senate of Pakistan adopted a resolution against Charlie Hebdo cartoons and shared this with the European Union's ambassador to Pakistan (Haider 2015). Following another occasion of a French magazine publishing blasphemous cartoons, Islamists came out on the streets of Pakistan demanding the government expel the French Ambassador in Pakistan (Baloch and Ellis-Petersen 2021). It is no surprise that the JI and JUI-F, being anti-Western, were at the forefront in such demonstrations in Pakistan. For instance, addressing a rally in Peshawar, JI's Ibrahim said,

> The Organization of the Islamic Cooperation should call an emergency meeting and take a collective stance over the issue. The European countries must be conveyed the message that the insult of Islam and Holy Prophet Muhammad (peace be upon him) will never be tolerated. (J. A. Khan 2015)

A more recent case of JI and JUI-F's anti-Westernism and a desire for a stricter form of Sharia was observed in their reactions following the Taliban's takeover in Afghanistan in August 2021. Like many other religious or right-wing leaders, for example, Maulana Abdul Aziz of Lal Masjid in Islamabad (Momand 2021), JI and JUI-F leaders have also praised the Taliban's victory against the US. Soon after the Taliban's takeover, JUI-F's leader Maulana Fazlur Rehman sent a congratulatory message to the Islamic Emirate of Afghanistan, and the JI expressed pleasure over the Taliban's remarkable victory (S. Khan 2021). By examining such reactions from a variety of segments in Pakistan, especially Islamists, Abbas and Ahmed (2021) argue that the reactions of the JUI and JI are linked

to their involvement in the Afghan–Soviet War as "they have been trying to equate the Taliban's victory with Fatah-e-Mecca, the conquest of Mecca, where Prophet Muhammad entered peacefully into Mecca". For the JI, often anti-Americanism or anti-Westernism and secularism are interlinked because their leaders tackle both issues together in their speeches. Sirajul Haq of the JI, for example, in August 2021, addressed a gathering in which he expressed that a strong Islamic government in Afghanistan will inspire the Muslim world. After blaming the US for killing innocent Muslims globally, Haq claimed that his party would not allow any conspiracy to turn Pakistan into a secular state (Azikhel 2021). Haq has been very vocal against the US and has been labeling the country as an enemy of Pakistan, Islam, and religious parties in the country. Earlier, following the 2018 general elections in Pakistan, Haq addressed a gathering in KP with the following remarks:

> The US was saying it was glad over the defeat of the religious parties in this country and that the Pakistanis have rejected extremism and terrorism. In fact, the US itself was promoting terrorism and extremism in the world ... People were always ready to lay down their lives for Islam and the Holy Prophet and no one can dare to convert this country into a secular state. (Dawn 2018)

The prominent non-religious political parties are somewhat similar in relation to anti-Westernism, albeit at a limited scale, but have tried to promote reforms and projects on minority rights in Pakistan. Based on his anti-Western and pro-Taliban rhetoric, Imran Khan of the PTI has been viewed as both the left and right-wing (Afzal 2019).

Thinking of anti-Westernism, one cannot ignore the era under Zulfiqar Ali Bhutto of the PPP. His politics shifted from left-wing as he began using Islam, pan-Islamism, and anti-Westernism in Pakistan, which was facing new challenges or realities following its disintegration in 1971. Following the creation of Bangladesh in East Pakistan, Bhutto led Pakistan away from Western security alliances that he had deemed useless to address Pakistan's security needs vis-à-vis India, as the US had offered no help to Pakistan during its 1971 war with India. Bhutto turned to the ummah after breaking with the West to advance Pakistan's national goals, namely, economic growth and security. Naturally, this sparked a resurgence of pan-Islamism in Pakistan (Ahmed and Akbarzadeh 2023). Bhutto's goal was to strengthen pan-Islamism by developing close ties with Muslim nations. In January 1972, he travelled to Afghanistan, Algeria, Egypt, Iran, Libya, Morocco, Syria, Tunisia, and Turkey in order to promote that agenda (Rizvi 1993). Bhutto's pan-Islamic foreign policy sought to lessen reliance on the US financially by enlisting the aid of wealthy Muslim nations such as Libya and Saudi Arabia. He was effective in fostering ties with important Muslim nations, for instance, by planning the second Organization of Islamic Cooperation (OIC) conference in Lahore in 1974, where Colonel Gadhafi called Pakistan "the citadel of Islam in Asia" (Bhutto 2010, p. 111).

The PML-N, or its top leadership in this regard, cannot be labelled anti-West. In fact, Nawaz Sharif is currently living with his sons, who are British citizens, in London. The party, however, has been carefully treading in Pakistan's political landscape, in which Islam has prominence. This is evident as the PML-N's leadership has never spoken against reforms that could pave the way for secularism or more minority rights in Pakistan, for example, changing the blasphemy laws. In 2020, Nawaz Sharif ordered the removal of blasphemous content from social media (Reuters 2017).

### 2.3. Religious Minorities

The Islamists narrative on anti-secularism is linked to how they view minority rights in Pakistan. While promoting Sharia, Islamists have frequently criticized what they deem Western-motivated secularism under the disguise of minority rights. This resistance is led by prominent religious parties, specifically the JI and JUI. This is even though minority rights are part of their election manifestos (see examples below).

> Minorities will enjoy rights to education, employment and other civil liberties ...
> Any discrimination, injustice or bias towards minorities will be discouraged. (JI 2013, p. 31)

> Minorities will be equal citizens of Pakistan and they will have all freedoms
> guaranteed in Islam and the constitution . . . Non-Muslim minority in the country
> will enjoy religious freedom, civil rights, and impartial opportunity to access to
> justice. (JUI-F 2013, p. 11)

Talking about minority rights is one thing, but believing in religious freedom is another, as that would require some fundamental changes in the constitution of Pakistan. The two most prominent religious parties, namely, the JI and the JUI, have shown through their actions that they are against secularism and minority rights. Both parties were successful in opposing reforms in the past, subsequently, rolling them back, which accounts for their inflexibility. The Sindh Assembly approved a bill in November 2016 that forbade the forced conversion of anybody under the age of 18 to any religion. All religious parties, especially the JI and JUI, strongly opposed this by considering it un-Islamic. The JI pressurized the PPP-led government in Sindh, which eventually gave in by withdrawing the bill (Ghori 2016). Former Pakistani minister Javed Jabbar remarked, "Religiosity has also bred murderous extremism, gravely damaging the nation internally, and totally distorting its image", after praising the Sindh Assembly's Act in response to criticism from religious parties (Jabbar 2016). In addition to maintaining their positions regarding the supremacy of Islam/Muslims in Pakistan, the two parties have never been seen publicly condemning attacks on people of other religions and their worship places. In fact, the JUI-F was found responsible for a major attack in 2020 on the Shri Paramhans Ji Maharaj Samadhi, a Hindu temple, in KP. More than 300 persons were booked for this attack, including JUI-F's district leader Maulana Mirza Aqeem (Chaudhry 2020).

When it comes to the issue of minority rights, most non-religious parties have very different positions. While Imran Khan was blamed for defending Pakistan's blasphemy laws, leaders of the PML-N and PPP have repeatedly talked about the misuse of these laws in Pakistan. In 2017, Nawaz Sharif (then Prime Minister) accepted that Pakistan's minorities are "unjustly treated" (Reuters 2017). From the PPP, its current chairman Bilawal Bhutto Zardari has been vocal regarding the misuse of blasphemy laws and the need to change these laws (Hussain 2011). Still, neither the PPP nor the PML-N made any such move under their governments. The PTI was accused of opposing religious forms in the shape of changing the blasphemy laws to forge stronger alliances with religious parties such as the JI. Leading up to the 2018 general elections, Imran Khan said at a gathering in Islamabad, "We are standing with Article 295c and will defend it" (Barker 2018). The Article 295c of the Pakistan Penal Code allows the death penalty or life imprisonment for the criminal offence of defiling the name of the Prophet Muhammad (Amnesty 2021).

*2.4. Secularism*

Pakistan's journey from secularism to theocracy is a clear sign of the Islamists' victory. While initially against the Two-Nation Theory or the idea of a separate homeland for the Muslims in the shape of Pakistan, for example, Maulana Maududi of the JI was skeptical about the nature of a new state under secular leaders such as Jinnah, and the Ulema such as Maulana Abul Kalam Azad (Jamiat Ulema-e-Hind) were less interested in the creation of a Muslim state (Jaffrelot 2012). Islamists were quick to recalibrate their strategies after 1947. Soon after the creation of Pakistan, the JI began supporting the creation of an Islamic state. Then, the JI's chief, Maududi, and more than other 30 Islamic scholars began deliberating on Pakistan's first constitution. In fact, Maududi was invited by Jinnah to deliver some lectures on the foundations of the Islamic system and governance (Qazi 2017).

The JI has been engaged in the process of defining an Islamic state. Hence, the party and its founder, Maulana Maududi, headed demonstrations to declare Ahmadiyya a non-Muslim minority in Pakistan. Surbuland (2022, p. 3) argues, "Maududi sought to articulate an Islamic politics that was exclusionist in nature". In 2002, JI's chief at the time, Munawar Hussain, said that his party would continue its struggle against secularism as Pakistan was created for Islam, not secularism (Nawai-i-Waqt 2002). Despite negligible electoral success, Islamists have maintained some influence in policy-making in some selected areas,

such as Islamic law. The Council of Islamic Ideology is completely under the direction of Islamists, demonstrating their influence in matters of religion, national identity, and religion. Looking at the Musharraf era, the MMA (a coalition of religious parties) wanted to implement Sharia in its true form in the country. MMA was formed before the 2002 general elections. While promoting Islamization, JUI's chief Maulana Fazl-ur-Rahman said that "true Islamization" was required to address the victimization of religious parties (K. K. Shahid 2018).

The demands of the Islamists have been met by successive governments, but it is crucial to note that non-religious parties and their governments and even military regimes have taken steps that led to the state's involvement in religious affairs. There were regimes, for example, of Ayub Khan and Zulfiqar Ali Bhutto, that claimed to be secular, but under pressure from Islamists, such as the JI, they paved the way for Islamization, which can be seen as signs of those governments managing religion. This does not mean that Khan and Bhutto were fully behind certain religious and constitutional reforms, but they did so for their survival. A major wave of Islamization happened under a left-leaning so-called secular party, the PPP, and a clear example of that is Pakistan becoming a theocracy through the 1973 constitution (Hamdani 2022).

Scholars have talked about competing ideas concerning the state–religion relationship and diverse understandings of secularism in Pakistan (Surbuland 2022; Ahmar 2012). They have, however, ignored a gap between rhetoric and reality, as even the non-religious parties are reluctant to either openly advocate for the separation of religion from state affairs or address barriers to secularism, such as the blasphemy laws, in Pakistan. The PPP's case under Zulfiqar Ali Bhutto was different as he was using religious labels in domestic politics, such as Islamic socialism, and foreign policy through a tilt towards pan-Islamism (N. N. Paracha 2013). Bhutto, however, never claimed to be a pious Muslim, unlike Imran Khan. While Khan has spoken of minority rights in Pakistan, he is also fearful of being labelled a secular. This could be because of various reasons, such as other politicians often labelling him as a Western/Zionist agent (Yasmeen 2013).

There remains confusion in Pakistan regarding secularism, and this is created by the ones who see secularism as anti-Islam. This could also be, as Ahmar (2012, p. 219) argues, due to the lack of awareness as the "Urdu translation of secular is 'atheist', which makes it very difficult for an ordinary Muslim Pakistani to resist and challenge the misinterpretation of secular and secularism". In response to allegations of promoting secularism, the PTI published an article on its website in its defense. In this blog, the author justified how Imran Khan is not secular as he is someone who talks about "Riasat-e-Medina", justice, and welfare in Pakistan (Mumtaz 2018). It seems that in the Islamic Republic of Pakistan, no politicians would like to carry the secular label.

Looking closely at the election manifestos of selected non-religious parties, it can be clearly seen that they are more Islamic than religious parties in terms of the promotion of Islam and Islamic Studies in Pakistan. This is particularly the case of the PML-N's 2018 election manifesto, in which under "religious affairs: peace and tolerance", the party carefully crafted its position on the issue of minority rights:

> PML(N) wants to make Pakistan a peaceful and progressive country wherein people can live their lives in accordance with the teachings and requirements of Islam as set out in the Holy Quran and Sunnah; wherein adequate provisions shall be made for minorities to freely profess and practice their religions and develop their cultures; wherein citizens shall be guaranteed fundamental rights of freedom of thought, expression, belief, faith, worship and association; and wherein adequate provision shall be made to safeguard the legitimate interests of minorities. (PML-N 2018)

While the PPP has a history of using religion in domestic politics and foreign policy, especially under the era of Zulfiqar Ali Bhutto, its election manifestos have used words that are absent from the election manifestos of the PML-N and PTI. This could be because Bhutto was still a secular despite giving in to the demands of Islamists—a sign of his

government managing religious matters in Pakistan. Here, noticeably, is the promotion of religious harmony and tolerance to prevent religiously motivated violence in Pakistan. The party's 2018 election manifesto also talked about taking action to stop the forceful conversion of religious minorities (PPP 2018, p. 49). Such issues are also context-specific to Sindh where the PPP dominates and where a Hindu minority faces forceful religious conversion. More than 90 percent of Pakistan's Hindus live in Sindh (Singh 2019), making the community a significant vote bank for the PPP. This was also evident in the outcomes of the 2018 generation elections when PPP's Mahesh Malani (a Hindu) from Sindh became the first non-Muslim parliamentarian after 2002 (Samoon 2018). While not mentioning the word secularism, the party advocated for the rights of religious minorities and pledged its struggle for an egalitarian Pakistan (Zardari 2020; PPP 2020).

The youngest party in comparison to the other chosen in this paper, the PTI, is more engaged in discussions on Pakistan's identity and raison d'etre. Hence, its 2018 election manifesto started with the following quote from the founding father of Pakistan, Muhammad Ali Jinnah: "We should have a State in which we could live and breathe as free men and . . . where principles of Islamic social justice could find free play" (PTI 2018, p. 4). The theme of justice is central to the party's vision and mission, as is also reflected in this name, where the word "insaf" means justice. Despite the secular leanings of Imran Khan—noticeable through some actions under the PTI government, such as declaring the Panj Tirath Hindu religious site as a national heritage and the opening of the Kartarpur Corridor for Sikh pilgrims from India (Z. S. Ahmed 2021)—there was no mention of the party aiming for a secular Pakistan.

Non-religious parties are also religious in many ways. First, none of them have ever claimed that they are secular or that they support the separation of state and religion in Pakistan. Second, non-religious parties also understand that there are certain no-go areas if they need to survive in the political and social landscape of Pakistan, as Islamists would react strongly to any such reforms as they have almost always done in the past. The blasphemy laws in Pakistan have been criticized for their misuse in terms of targeting religious minorities. It has, however, been a sensitive topic, as speaking against these laws can have serious consequences. In 2011, Punjab's then-Governor, Salman Taseer, and Pakistan Minorities Minister Shahbaz Bhatti were assassinated for speaking against Pakistan's blasphemy laws. While religious parties supported the murderer of Taseer, leaders of non-religious parties remained largely silent (Bajoria 2011). The head of PML-N, however, condemned the attack but also said that Taseer should have "adopted a more balanced approach" (Talwar 2011).

Most recently, the PTI government walked on this path. For the party's leader, Imran Khan, who was labelled a Western/Zionist agent (Yasmeen 2013) and secular, it was not easy to initiate certain reforms. However, the party has never claimed to be secular and, in fact, has promoted greater involvement of Islam or true Islam, as Imran Khan says, in every sphere of life. During the PTI's government, the party acted like a religious party in various ways. For example, its government in Punjab made the teaching of the Quran compulsory in schools and the party boasted on its website: "A historic step taken by the PTI Government—for . . . a nation that was created in the name of Islam to understand and follow the spirit and ethos of the faith holds a central place" (PTI 2020). On the issue of a single national curriculum, the party faced a backlash from Islamists; for example, the JI's chief felt that a single national curriculum was an attempt to secularize Pakistan by removing Islamic content from subjects such as Pakistan Studies, Urdu, and History (Business Recorder 2021). In response, the PTI government changed the name of its proposed curriculum by dropping the world "single" to just the National Curriculum of Pakistan (Dawn 2022). In 2021, Khan opened Al-Qadir University in Jhelum for Sufi teachings and on the occasion said,

> Islam and science were both working alongside each other . . . What is the purpose of Al-Qadir University, how can our religion become relevant in the lives of our people. The [university] will develop the minds in Pakistan. Right now youth is

highly influenced by the West . . . but as humans are they [youth] growing in the right way, are they going down the right path? (Arab News 2021)

It is important to make sense of why the Islamists oppose secularism or secularization in Pakistan. To understand, one needs to look at both the JI and JUI, which are products of Islamic revivalist movements during the colonial era. During the British colonial rule in the Indian sub-continent, Islamists focused on Islamic fundamentalism. The current positions of Islamists regarding secularism in Pakistan have not changed since the colonial era. As argued by Iqtidar (2012, p. 57), "The Islamists are vehement in their public insistence on dislodging the idea of secularism as universal, claiming it to be a parochial, European experience—with some justification". As was the case of JI's Maududi, the current leadership of the party also looks at modernism in connection with Europeanization or Westernization. Hence, Maududi's portrayal of "divine sovereignty" was seen as a reaction of modernity as he "refused to accept colonial epistemic hegemony" (Iqtidar and Scharbrodt 2022, p. 278). While founding fathers of JUI had accepted secular democracy in the light of Qur'an and Sunnah, Maududi rejected secular democracy by calling it *taghuti nizam* (satanic system) (Engineer 2010). As far as the matter of the Islamists' positions on secularism are concerned, they have not changed since the colonia era or the time of Islamic revivalist movements in the Indian sub-continent.

As far as the selected non-religious political parties are concerned, they are not similar in terms of their ideologies. The PPP is center-left, the PTI right-of-center, and the PML-N center-right. Knowing this distinction is important in terms of understanding their positions on secularism and secularization in Pakistan. Seen as a secular man, the founding father of the PPP, Zulfiqar Ali Bhutto, took many steps that advanced the entanglement of state with religion (Chacko 2016). The party's interim constitution stated, "Islam is our faith" (Chengappa 2002, p. 29). While the party's leaders, including Benazir Bhutto, have made references to Islam in their speeches (Laila et al. 2020), the PPP stands out in terms of promoting secularization in the country by speaking on issues such as the blasphemy laws and minority rights. This is in line with the party's center-left ideology, unlike the PTI and PML-N, which have leaned towards Islam.

## 3. Conclusions

Pakistan was created based on the Two-Nation Theory as a nation-state, and this theory or the vision continues to serve as a lens through which Pakistan's identity is perceived and shaped by major political parties. The analysis shows Islamists have very rigid positions on issues concerning secularism, modernity, and secularism in Pakistan. As is clear from the analysis of the JI and JUI-F, one can see that the parties' current positions on these matters are no different from the founding fathers of these parties, such as Maulana Maududi and Mufti Mahmood, who were openly against secularism. While non-religious parties have used religion in domestic politics and foreign affairs, they have demonstrated different trajectories on issues covered in this paper. Often for the sake of their political survival, civil and military elites made concessions to Islamists, and this was evident during the earlier decades following the creation of Pakistan under Ayub Khan and Zulfiqar Ali Bhutto. Both were secular but managed religious affairs, involving the state more in religious practice and Islamic thought, similar to the case of General Musharraf's "Enlightened Moderation". The trend of Islamists acting as pressure groups against certain reforms has continued as almost all ruling parties had to backtrack on proposed policy reforms, for example, the Sindh government's proposal to ban the forced conversion of religious minorities and the PTI's proposal on a single national curriculum.

The analysis of five political parties in this paper shows that there is no opposition to the idea that Pakistan was created based on the Two-Nation Theory, and therefore, it is not a secular state. Among the parties, the PPP is the only party that claims to be secular or whose key leaders are viewed as secular. In the case of Islamists, it is clear why they refrained from using the term "secularism" as, for example, their leaders have a long history of viewing secularism as a Western conspiracy and incompatible with Islam. While

the religious parties have long been blaming the West for the Muslim world's downfall, anti-Western sentiments were also promoted by Zulfiqar Ali Bhutto of the PPP, and the case of Imran Khan is similar.

Islamists resist any changes to what they view as their hard-fought success, for example, through Islamization. This is evident through their opposition to changing the blasphemy laws, but leaders of PML-N and PTI have also defended the country's blasphemy laws. The PPP is an exception in this case because its leaders have most often talked about the misuse of blasphemy laws against religious minorities. The PPP, when in government, made no move to change the blasphemy laws. Moreover, like other parties examined in this study, the PPP has never claimed to be secular but to be the ones that also believe in minority rights and, most of all, in its egalitarianism. Unlike the PPP in the past, for example, under Zulfiqar Ali Bhutto and today's PTI, PML-N was not publicly advocating for a model Islamic welfare state through ideas such as Islamic socialism and "Riasat-e-Medina", but like other political parties, it has also never aimed for the separation of religion and the state. It seems that the status quo suits the political leadership of the chosen parties or today's political elites as that maintains majoritarianism in Pakistan.

**Funding:** This research received no external funding.

**Conflicts of Interest:** The author declares no conflict of interest.

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
