# Peer review of "Islam and the Politics of Secularism in Pakistan"

_religions, doi:10.3390/rel14030416_

Round 1
Reviewer 1 Report
The basic argument of the article is important and well-presented. Going back to Zulfiqar Bhutto (and further), the weak position of religious minorities in Pakistan has been due not only to the Islamic parties but to the policies and positions of parties and politicians often viewed as more secular as well. The article makes this important point clearly and with good evidence, and provides some analysis of why this is so (though this analysis could certainly be pressed further with more analysis of the structure of Pakistani politics -- beyond the throwaway final sentence on the attraction of current elites to "majoritarianism"). Having said that, its use of "secularism" as an analytical category does not do justice to the complexity of the literature on this subject. I was also a little surprised by the lack of discussion of the term "Islamist," which is commonly applied to the JI, but somewhat less commonly applied to ulama-based parties like the JUI. Overall, the article is valuable enough to be published, in my view, but offers only a limited contribution to the larger literature on secularism. If that is the aim, it could use some revisions with this in mind.
Author Response
Response: I thank the reviewer for these encouraging remarks. I have taken care of all the suggestions by firstly problematizing secularism in the introduction and using that literature, especially the one that highlights the relationship between Islam and secularism in the analysis. I have also added an explanation of how the JUI is categorized under the term “Islam”. It is mainly because the party’s political mission is to advocate for Islamic fundamentalism and a greater application of Islamic law in Pakistan. In this regard, it is similar to the JI.
Reviewer 2 Report
I read this article with great interest. The article is generally clearly written. There is a kernel of an interesting argument here. It will, however, need significant development to bring that out more clearly. Here are some suggestions for improvement that I hope the author will find useful.
Argument and Evidence:
The argument, as I understand it, is that Pakistan's so called 'non-religious' parties have not supported secularism and their discourse is very similar to the religious parties. Crucial to the author's argument here is the definition of secularism. The sources they cite such as Iqtidar and Gilmartin actually work with a much more sophisticated definition of secularism, not as the separation of religion and state as the author contends, but as the management of religion by the state. There is an important distinction and the author has a) miscited and b) failed to engage with a definition that would yield more productive results for this analysis. Working with this more complex definition would help them assess why parties like PTI, PPP and PML continue to rely upon Islamic framing to legitimize their policies. Currently the article describes but does not analyse.
To this end, the author needs to clarify the value of appropriating secularism by political parties. Suppose the parties did officially support secularism, what difference would this make? This is related to the problem of evidence. The evidence the author provides requires strengthening. The author has skipped the early years of Pakistan's formation when secularism was officially proclaimed by military dictatorships such Ayub Khan's as well as democratically elected parties such as PPP. Indeed, they may wish to look at Ali Usman Qasmi's article in Modern Asian Studies about the process through which the self-proclaimed secular PPP dominate parliament approved the change in status of Ahmadi muslims to non-muslim status.
At the core of their argument then there is a big gap about why political parties are unwilling to proclaim or support secularism as separation of religion and state? Might it be possible that they feel they are better able to manage religion through appropriating it but using it for their own purposes? If that is the case, then at least part of the problem is not with their support for secularism or lack of it, but the associations within their constituencies with it.
Methodology
Part of the author's claim is that they are using CDA to assess the place of secularism in political discourse and that has not been done before. However, the article does not really draw upon the 'critical' aspect of critical discourse analysis. That would require deeper historical contextualization, questioning of their own assumptions about key terms and deeper engagement with existing scholarship on this including Asad Ahmed's work on blasphemy laws in Pakistan, Humeira Iqtidar's work on the relationship between secularism and secularization, and Ali Qasmi's work on religious opposition to Pakistan.
Structure
The structure needs some further consideration. I am not persuaded by the author's decision to first provide an overview of religious parties statements (which it has to be noted are different from discourse, discourse needs more development of also the unsaid assumptions and implications) and then of other parties. The two sections with the same sub-sections (secularism, minorities etc.) feel repetitive and take up word space that would have been better utilized providing an analysis rather than a description. The two sections can be combined and within each sub-section, e.g. minority rights, the author can compare the positions of both types of parties.
Author Response
Reviewer 2: I read this article with great interest. The article is generally clearly written. There is a kernel of an interesting argument here. It will, however, need significant development to bring that out more clearly. Here are some suggestions for improvement that I hope the author will find useful.
Response: I thank the reviewer for having some faith in the paper. I have accordingly revised it.
Reviewer 2: The argument, as I understand it, is that Pakistan's so-called 'non-religious' parties have not supported secularism and their discourse is very similar to the religious parties. Crucial to the author's argument here is the definition of secularism. The sources they cite such as Iqtidar and Gilmartin actually work with a much more sophisticated definition of secularism, not as the separation of religion and state as the author contends, but as the management of religion by the state. There is an important distinction and the author has a) miscited and b) failed to engage with a definition that would yield more productive results for this analysis. Working with this more complex definition would help them assess why parties like PTI, PPP and PML continue to rely upon Islamic framing to legitimize their policies. Currently the article describes but does not analyse.
Response: I agree with these observations and have accordingly expanded the discussion on secularism by looking at the work of Talal Asad vis-à-vis management of religion by the state. This definition is particularly relevant to the case of how various religious parties have managed religion in Pakistan.
Reviewer 2: To this end, the author needs to clarify the value of appropriating secularism by political parties. Suppose the parties did officially support secularism, what difference would this make? This is related to the problem of evidence. The evidence the author provides requires strengthening. The author has skipped the early years of Pakistan's formation when secularism was officially proclaimed by military dictatorships such Ayub Khan's as well as democratically elected parties such as PPP. Indeed, they may wish to look at Ali Usman Qasmi's article in Modern Asian Studies about the process through which the self-proclaimed secular PPP dominate parliament approved the change in status of Ahmadi muslims to non-muslim status.
Response: These are valid points and accordingly I have revised them by discussing secularism under Ayub Khan and Bhutto and how the two regimes faced challenges and tried to manage religion.
Reviewer 2: At the core of their argument then there is a big gap about why political parties are unwilling to proclaim or support secularism as separation of religion and state? Might it be possible that they feel they are better able to manage religion through appropriating it but using it for their own purposes? If that is the case, then at least part of the problem is not with their support for secularism or lack of it, but the associations within their constituencies with it.
Response: This is a valid point and might be true for almost all military authoritarian rules and the PPP under Zulfiqar Ali Bhutto. This however cannot be said for the Islamists and the parties that center-right with a clear religious agenda, for example in the case of the PTI and PML-N. I have found these comments very useful and accordingly revised analysis by differentiating between the selected political parties.
Reviewer 2: Part of the author's claim is that they are using CDA to assess the place of secularism in political discourse and that has not been done before. However, the article does not really draw upon the 'critical' aspect of critical discourse analysis. That would require deeper historical contextualization, questioning of their own assumptions about key terms and deeper engagement with existing scholarship on this including Asad Ahmed's work on blasphemy laws in Pakistan, Humeira Iqtidar's work on the relationship between secularism and secularization, and Ali Qasmi's work on religious opposition to Pakistan.
Response: Thanks for these comments as I went through several high-quality papers of Iqtidar on relevant issues. It was very interesting to see how she differentiated secularism and secularization. The inclusion of such arguments has helped in terms of expanding CDA in this paper. For the greater application of CDA, I have looked at the historical positions and political ideologies of selected political parties to expand the analysis in this paper. I am thankful for these suggestions.
Reviewer 2: The structure needs some further consideration. I am not persuaded by the author's decision to first provide an overview of religious parties statements (which it has to be noted are different from discourse, discourse needs more development of also the unsaid assumptions and implications) and then of other parties. The two sections with the same sub-sections (secularism, minorities etc.) feel repetitive and take up word space that would have been better utilized providing an analysis rather than a description. The two sections can be combined and within each sub-section, e.g. minority rights, the author can compare the positions of both types of parties.
Response: I agree and accordingly have restructured the paper. This now looks a lot better with various thematic sections in which the positions of Islamists and non-religious parties are compared. I have also strengthened the analysis by digging deeper into the positions of the political parties covered in this analysis.